# Deep Learning Approach for Differentiating Etiologies of Pediatric Retinal Hemorrhages: A Multicenter Study

**DOI:** 10.3390/ijms242015105

**Published:** 2023-10-12

**Authors:** Pooya Khosravi, Nolan A. Huck, Kourosh Shahraki, Stephen C. Hunter, Clifford Neil Danza, So Young Kim, Brian J. Forbes, Shuan Dai, Alex V. Levin, Gil Binenbaum, Peter D. Chang, Donny W. Suh

**Affiliations:** 1Department of Ophthalmology, School of Medicine, University of California, Irvine, CA 92697, USA; pooyak@hs.uci.edu (P.K.); nahuck@hs.uci.edu (N.A.H.); kourosh.shahyar@gmail.com (K.S.); cdanza@hs.uci.edu (C.N.D.); 2Gavin Herbert Eye Institute, University of California, Irvine, CA 92697, USA; 3Donald Bren School of Information and Computer Sciences, University of California, Irvine, CA 92697, USA; changp6@hs.uci.edu; 4School of Medicine, University of California, 900 University Ave, Riverside, CA 92521, USA; stephen.hunter@medsch.ucr.edu; 5Department of Ophthalmology, College of Medicine, Soonchunhyang University, Cheonan 31151, Chungcheongnam-do, Republic of Korea; ophdrkim@gmail.com; 6Division of Ophthalmology, Children’s Hospital of Philadelphia, Philadelphia, PA 19104, USA; forbesb@email.chop.edu (B.J.F.); binenbaum@chop.edu (G.B.); 7Department of Ophthalmology, Queensland Children’s Hospital, South Brisbane, QLD 4101, Australia; shuan.dai@health.qld.gov.au; 8Department of Ophthalmology, Flaum Eye Institute, Golisano Children’s Hospital, Rochester, NY 14642, USA; alex_levin@urmc.rochester.edu; 9Department of Radiological Sciences, School of Medicine, University of California, Irvine, CA 92697, USA

**Keywords:** artificial intelligence, deep learning, pediatrics, retinal hemorrhage

## Abstract

Retinal hemorrhages in pediatric patients can be a diagnostic challenge for ophthalmologists. These hemorrhages can occur due to various underlying etiologies, including abusive head trauma, accidental trauma, and medical conditions. Accurate identification of the etiology is crucial for appropriate management and legal considerations. In recent years, deep learning techniques have shown promise in assisting healthcare professionals in making more accurate and timely diagnosis of a variety of disorders. We explore the potential of deep learning approaches for differentiating etiologies of pediatric retinal hemorrhages. Our study, which spanned multiple centers, analyzed 898 images, resulting in a final dataset of 597 retinal hemorrhage fundus photos categorized into medical (49.9%) and trauma (50.1%) etiologies. Deep learning models, specifically those based on ResNet and transformer architectures, were applied; FastViT-SA12, a hybrid transformer model, achieved the highest accuracy (90.55%) and area under the receiver operating characteristic curve (AUC) of 90.55%, while ResNet18 secured the highest sensitivity value (96.77%) on an independent test dataset. The study highlighted areas for optimization in artificial intelligence (AI) models specifically for pediatric retinal hemorrhages. While AI proves valuable in diagnosing these hemorrhages, the expertise of medical professionals remains irreplaceable. Collaborative efforts between AI specialists and pediatric ophthalmologists are crucial to fully harness AI’s potential in diagnosing etiologies of pediatric retinal hemorrhages.

## 1. Introduction

Retinal hemorrhages can pose substantial diagnostic challenges due to their association with various systemic and ocular diseases [1]. The retina comprises a total of 10 layers, with the outermost layer being the retinal pigmented epithelium (RPE), followed by 9 other layers collectively known as the neurosensory retina. The innermost layer is referred to as the inner limiting membrane (ILM) [2]. Hemorrhages can occur within the layers of the retina (intraretinal) or above the retina itself (preretinal). Subretinal hemorrhages are an accumulation of blood between the neurosensory retina and the RPE. If the blood extends into the vitreous humor, it is referred to as vitreous hemorrhage. Intraretinal hemorrhages that occur in the superficial layers of the retina exhibit a linear streaking appearance, following the pattern of the nerve fibers, often referred to as flame or splinter hemorrhages due to their distinct appearance. On the other hand, intraretinal hemorrhages that occur in the deeper layers of the retina have a more rounded shape and are referred to as dot or blot hemorrhages, depending on their size [2,3,4].

The differential diagnosis of retinal hemorrhages is extensive and includes [3,5] coagulopathy, blood dyscrasia, severe anemia, hemolytic uremic syndrome, endocarditis, vasculitis [2,3], normal birth [6], raised intracranial pressure (ICP), glutaric aciduria type 1, meningitis and Terson syndrome [7]. Retinal hemorrhages can also be diagnostic indicators for severe conditions such as abusive head trauma [1,2,8,9,10,11]. Common causes of retinal hemorrhage in adults include diabetic retinopathy, hypertensive retinopathy, retinal vein occlusion [2,3,12], trauma [9,13], and hematological disorders [2].

Fundus photography increases our ability to visualize, document, and monitor retinal hemorrhages [14,15]. While the interpretation of these images heavily relies on the trained eye of an experienced ophthalmologist, there is potential for further improving interpretation and consistency. This is where artificial intelligence (AI) and machine learning (ML) technologies can enhance ophthalmologic practice, offering advanced computational tools that complement, rather than replace, the expert judgment of clinical professionals. This helps reduce variability and ensure a more standardized approach across observers [15]. By doing so, AI and ML contribute to the ongoing development of improved diagnostic capabilities in ophthalmology.

Previous research has demonstrated the use of AI and ML to analyze fundus photographs [14,15,16]. These papers primarily focused on disease detection [17,18], classification and grading [18,19], segmentation [20,21] and prediction [22] for retinopathy of prematurity (ROP) [23,24], diabetic retinopathy [18,25,26], age-related macular degeneration [21,27,28,29,30], glaucoma [31,32,33], cataracts [34], lacrimal disorders [35], keratoconus [36], amblyopia [37] and optic nerve diseases [38,39]. Studies have demonstrated that AI and ML algorithms, including deep learning models such as convolutional neural networks (CNNs), RNNs, and Transformers, can accurately identify and classify different types of retinal lesions, such as hemorrhages, exudates, and micro-aneurysms [25,26]. These algorithms can analyze large datasets of fundus photographs and provide automated assessments of disease severity and progression. Furthermore, AI and ML have been utilized to predict the risk of developing certain retinal diseases, such as geographic atrophy, based on fundus image analysis [27,29]. By training algorithms on large datasets of fundus photographs and corresponding patient data, these technologies can identify patterns and markers that are indicative of disease progression or future complications [29].

Unlike previous studies that primarily aimed to describe abnormal exams, our study seeks to develop and validate CNN and transformer models to differentiate between patients with known retinal hemorrhage. This is an especially challenging task as retinal hemorrhages can be difficult to interpret even for trained experts. To date, despite the progress made in applying AI in ophthalmology, there is still a lack of research exploring the potential of computational tools in differentiating the etiologies of retinal hemorrhages.

## 2. Results

Our study spanned multiple centers and involved a comprehensive analysis of 898 retinal hemorrhage (RH) fundus images of which 301 images were excluded due to the absence of retinal hemorrhage, resulting in a final dataset comprising 597 images. The images were divided into medical (*n* = 298, 49.9%), and trauma etiologies (*n* = 299, 50.1%). The detailed distribution of the number of photos corresponding to each diagnosis is provided in Table 1. Figure 1 showcases representative examples of these diagnoses.

Upon completing the pre-processing stage, the dataset was divided at the patient level into distinct subsets for training, validation, and testing, representing 60%, 20%, and 20% of the total dataset, respectively. The training subset (*n* = 343; 172 medical, 171 trauma) was used to adapt the model’s parameters and to learn the underlying patterns in differentiating medical from traumatic retinal hemorrhages. The validation subset (*n* = 127; 61 medical, 66 trauma) played a crucial role in tuning hyperparameters and selecting the best-performing model, ensuring the chosen model was robust yet not overfitted to the training data. The testing subset (*n* = 127; 65 medical, 62 trauma) was then used to provide an unbiased evaluation of the final model’s performance in unseen data. Dividing the data at the patient level ensured that all images from the same patient were contained within the same subset, minimizing the risk of data leakage between the subsets and thus providing a more rigorous and trustworthy evaluation of the model’s performance. This approach aligns with best practices in machine learning and provides a sound basis for extrapolating the findings to a broader population. The performance metrics of the models based on the ResNet architecture were highly compelling. Among the ResNet models, the ResNet18 model exhibited the most remarkable results, with an AUC of 0.9506 and an accuracy of 88.98%. It also demonstrated the highest sensitivity of 96.77% and a PPV of 84.29%. The ResNet101 model achieved an AUC of 0.9449, while attaining the highest accuracy, specificity, and PPV of 89.76%, 90.77%, and 90.16%, respectively out of the five models. Comprehensive performance metrics are presented in Table 2. AUC graphs (Appendix A) and confusion matrices (Appendix A) are included in the Appendix A.

The FastViT-SA12 model with the highest accuracy misclassified 2 medical and 10 trauma cases, while the ResNet18 model with the highest sensitivity misclassified 8 medical and 6 trauma cases. ViT-Small with the lowest accuracy misclassified 12 medical and 14 trauma cases. The misclassifications by diagnosis are presented in Appendix A. It should be noted that the errors made by all models exhibited a discernible pattern. Specifically, cases of acute myeloid leukemia (AML) and acute lymphoblastic leukemia (ALL), as well as other hematologic disorders and coagulopathies that caused significant retinal hemorrhaging, were often misclassified by all models as traumatic. This confusion likely arose due to the similarity in the number and extent of hemorrhaging between these medical conditions and trauma-related cases. Conversely, two instances of accidental and abusive head trauma, characterized by minimal retinal hemorrhage, were incorrectly identified as medical in origin. On the other hand, cases of papilledema and birth trauma were all classified correctly. Examples of these misclassifications are provided in Figure 2.

## 3. Discussion

The findings of this study illustrate the potential of deep learning models, particularly transformer models with attention mechanisms, in differentiating medical versus traumatic pediatric retinal hemorrhages (RH). This study is unique in that it focuses specifically on differentiating traumatic from medical retinal hemorrhages, rather than comparing normal and abnormal images. This distinction has significant medicolegal implications, particularly for cases involving abusive head trauma.

Although this is the first study that investigates distinguishing etiologies of retinal hemorrhages from fundus photos, there have been previous studies focused on detecting the presence, segmentation, or identifying the layer of retinal hemorrhages from fundus images [40,41,42,43].

The superior performance of FastVit-SA12 and ResNet18 in our experiments, as highlighted by the two highest AUCs among the evaluated models, underscores the importance of carefully selecting model architecture in accordance with the size of the available dataset. Both ResNet18’s and FastVit-SA12’s relatively streamlined architectures demonstrated a notable advantage in adapting to our limited dataset, potentially due to their smaller parameter count compared to the more intricate designs of larger ResNet and transformer models (Table 3). The relatively fewer parameters in FastVit-SA12 and ResNet18, in comparison to the larger models, may have conferred an advantage in curtailing overfitting on the limited data, consequently enhancing their ability to generalize effectively on both the validation set and the test set. Models characterized by heightened complexity, such as ResNet101, ResNet152, ViT-Small, and FastVit-SA24, while beneficial for larger datasets, might exhibit diminished efficiency on smaller datasets and increased susceptibility to overfitting to the training data. This aligns with the principle of Occam’s razor in model selection, which suggests that simpler models are often preferable when they perform as well, or better than, more complex models on the same task [44].

These results highlight the potential of transformers and CNNs as robust tools for distinguishing the medical versus traumatic RH in pediatric patients, even in scenarios involving relatively limited datasets. This finding is a testament to the need for striking a balance between model complexity and the risk of overfitting when working with limited datasets. Additional research using larger datasets would be advantageous in validating these findings and exploring the performance of more complex models.

The observed patterns of misclassification within our study provide valuable insights into areas for model optimization. Instances where extensive hemorrhages due to leukemia were identified as trauma, and a few accidental head trauma cases were predicted as medical. With an extensive RH, it may hide the underlying cotton wool spots, hard exudates that may help AI to make the correct diagnosis of medical induced RH. Also, if mild traumas are present in localized areas, AI may mistakenly interpret the image as sectoral RH. These findings present an opportunity for targeted refinements that could effectively enhance model performance. Furthermore, these subtle distinctions between medical and traumatic cases offer an avenue to deepen our comprehension of various pathophysiological presentations, potentially leading to the integration of supplementary clinical features or fine-tuned training strategies. Overall, the results accentuate the potential of the model to be a valuable diagnostic tool for differentiating retinal hemorrhages, while continually striving for enhancement and adaptation to the multifaceted nature of these conditions.

While our study aims to explore the potential of computational tools in differentiating the etiologies of retinal hemorrhages, it is important to acknowledge the expertise of experienced ophthalmologists as a benchmark in this field. The goal is to complement their expertise with AI and ML technologies, rather than replace it.

Despite the encouraging findings, our study is not without limitations. While our dataset encompasses a diverse range of RH etiologies, it is relatively small and constrained by factors such as photo quality, reliance on expert judgment, and historical chart information. The inconsistency in technologies employed to capture the images, coupled with the lack of standardization in luminance, magnification, and field, further contributes to these limitations. Additionally, some cases presented with mixed diagnoses, such as increased intracranial pressure with optic disc edema and trauma, adding complexity to classification. Our study also acted by proxy, with diagnoses predetermined based on medical history, laboratory and radiologic findings, physical examination, and multidisciplinary assessments. To address these limitations in future research, efforts should focus on acquiring a larger, higher-quality dataset that encompasses a more diverse set of retinal hemorrhage etiologies, while adhering to standardized image acquisition protocols to ensure consistency and reliability.

We conclude AI technology may be a helpful tool for assisting medical professionals in accurately distinguishing between medical and trauma-induced retinal hemorrhages. Of course, careful attention must always be paid to medical history, laboratory and radiologic findings, physical examination, and multidisciplinary assessments.

AI systems might not determine the precise biomechanism and degree of force responsible for the trauma that induced the RH. In cases where distinguishing between abusive and accidental physical trauma is crucial for legal or investigational purposes, the expertise of skilled medical professionals remains essential [10,11].

As the field of AI in medicine continues to advance, further research and refinement of algorithms may improve its capacity to gather additional contextual information and aid in unraveling the specific details surrounding traumatic events. Collaborations between AI experts and medical professionals will be pivotal in obtaining the full potential of AI while ensuring responsible and ethical implementation in the domain of medical diagnosis. Despite its current limitations, AI remains a promising and invaluable tool in the realm of healthcare, contributing significantly to improved diagnostic accuracy and patient care.

## 4. Methods and Materials

### 4.1. Study Population

Pediatric fundus photographs were collected from Soonchunhyang University Cheonan Hospital in Korea, Children’s Hospital of Philadelphia, Wills Eye Hospital, The Hospital for Sick Children (Toronto), Golisano Children’s Hospital and Flaum Eye Institute in Rochester, New York, NY, USA, and Queensland Children’s Hospital in Brisbane, Australia. The data collection period spanned from 2015 to 2023. These images were captured using either handheld portable devices (e.g., RetCam (RetCam 3, Natus Medical Inc., Pleasanton, CA, USA), ICON (Phoenix Clinical, Inc., Pleasanton, CA, USA), or Nonmyd7 (Kowa, Torrance, CA, USA)) or by upright standard fundus photography. We included cases that were confirmed abusive head trauma (AHT), a determination that was made through multidisciplinary child abuse team evaluation. We also included medically validated cases that were correlated with laboratory results as well as cases of accidental trauma as validated by witness, scene investigation and/or consistent physical findings. We excluded cases with uncertain etiology. The utilization of unrecognizable, anonymous images negated the requirement for Institutional Review Board (IRB) approval, given the absence of any identifiable personal information associated with the participants. Photographs were provided from the personal collections of the authors.

### 4.2. Annotation

Image labels were assigned based on the underlying case diagnosis. These labels were categorized as either traumatic or medical conditions based on medical records from the input of multi-specialists including the child abuse experts. The determination of hemorrhage presence or absence was entrusted to a panel of pediatric ophthalmologists (D.W.S., G.B., A.V.L., B.J.F.). In situations where there were varying interpretations of the image, a consensus was achieved through thoughtful deliberation until agreement was reached on the final classification.

### 4.3. Data Preprocessing

Prior to analysis, all images were subjected to standardized preprocessing procedures in order to ensure uniformity and enhance variability across images from different sources and devices. This involved resizing all images to a consistent shape of 256 × 256 pixels using bilinear interpolation, with zero-padding as necessary. Bilinear interpolation provided a reliable method for resizing images without introducing significant artifacts or distortions. Additionally, each image was normalized per channel based on the mean and standard deviation derived from the ImageNet training set [45]. During the training process, the augmentation steps included resizing the images to 224 × 224 to use transfer learning, applying random contrast ranging from 0.5 to 1.5, introducing Gaussian noise with a kernel size of 11 × 11 and sigma ranging from 0.1 to 2, randomly flipping the images horizontally with a 50% probability, and applying cropping with a scale range of 0.08 to 1. A minimum scale range of 0.08 ensures that even smaller details within the images are considered during training, potentially capturing subtle patterns that may be diagnostically relevant. Furthermore, an additional augmentation technique known as cutout [46], involving the placement of 4 randomly positioned obscuring boxes each measuring 44 × 44 (Figure 3), was employed. The comprehensive combination of these data augmentation methods aimed to prevent overfitting and promote the generalizability of the dataset. During the validation and testing phases, the images were resized to a matrix shape of 224 × 224 and underwent per-channel normalization, mirroring the preprocessing steps implemented during training.

### 4.4. Algorithm Development

In this study, our focus was the training of CNN and Transformer architectures to effectively discern between retinal hemorrhages of traumatic and medical origins. To achieve this, all experiments are derived from standard ResNet-based architectures [47] (ResNet34 shown in Figure 4), Residual Attention Networks (ResAttNet56) [48] using attention residual learning, a vision Transformer (ViT) image classification model [49], and FastViT [50], a hybrid vision transformer architecture (Table 3). Each model was fine-tuned after initialization with pretrained ImageNet weights from the PyTorch repository (torchvision 0.15.0; IMAGENET1K_V1) [51] for ResNet models and PyTorch Image Models [52,53] for ResAttNet56, ViT-small, and FastViT models.

For model fine-tuning, the default last fully connected layer is replaced with two custom fully connected hidden layers before projection to a two-element SoftMax activated logit score. The fully connected hidden layers are of dimensions 256 and 128, respectively, and employ a Rectified Linear Unit (ReLU) activation function. Additionally, the fully connected hidden layers are each implemented with batch normalization prior to ReLU activation and a dropout rate of 50%. During the fine-tuning phase, all layers of the network were unfrozen. This strategic approach enhances our models’ adaptability and enables them to effectively accommodate the unique characteristics of our dataset.

Each model was trained for a total of 50 epochs, except for ViT-Small, which was trained for 250 epochs. The model was optimized using Adam optimizer [54] with a standard binary cross-entropy loss function. We employed Hyperopt [55] for hyperparameter tuning and determined that a learning rate of 0.001 and a batch size of 32 consistently yielded the best performance across all ResNet and transformer models used in our study. Optimal model selection was determined based on validation data performance. Only after this stringent evaluation did we set forth to apply the chosen model to the test dataset for final assessment.

### 4.5. Statistics

For the purpose of this research, the dataset was randomly partitioned into training, validation, and test cohorts at a ratio of 60:20:20, respectively, on a per-patient basis (i.e., all images from the same patient were used in the same cohort). Rigorous evaluation of the model’s performance was conducted using the area under the receiver operating characteristic curve (AUC) on the independent test dataset. Furthermore, sensitivity, specificity, positive predictive values (PPV), and negative predictive values (NPV) were calculated at the Youden Index threshold, which effectively distinguishes cases with traumatic causes from medical causes. The Youden Index facilitates a comprehensive assessment of the ROC curve by pinpointing an optimal cutoff value that maximizes the difference between the true positive rate (sensitivity) and the false positive rate (1-specificity).

## Figures and Tables

**Figure 1 ijms-24-15105-f001:**
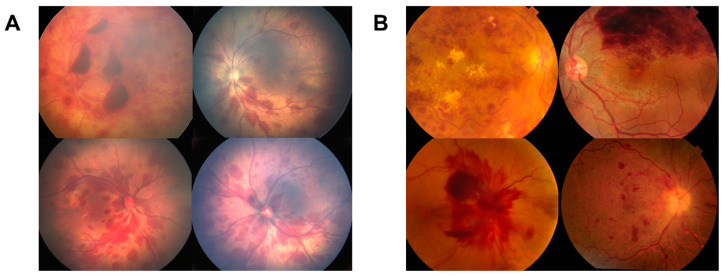
Representative fundus photographs from our pediatric dataset, categorized by diagnosis. This diverse collection includes images dichotomized into trauma cases (**A**) that encompass vaginal birth trauma (right), and abusive head trauma (left), and medical cases (**B**), such as ischemic CRVO (top left), coagulopathy (bottom left), BRVO (top right), and CRVO (bottom right).

**Figure 2 ijms-24-15105-f002:**
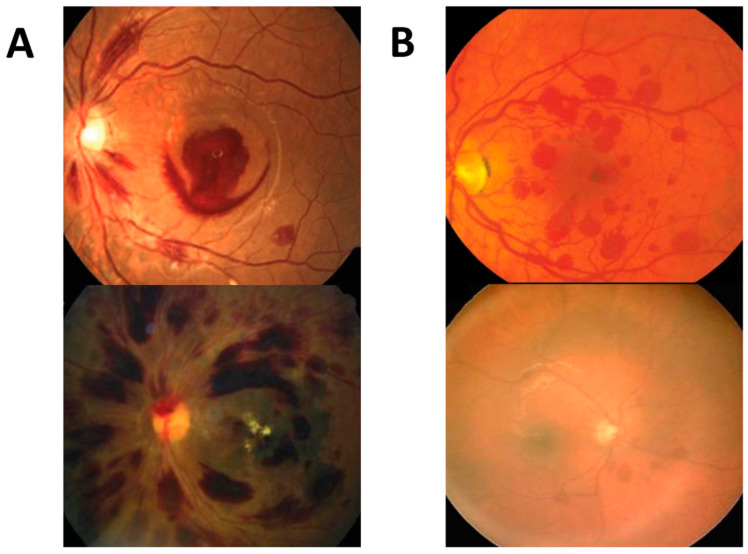
The two images on the left (**A**) illustrate medical cases (top left: leukemia, bottom left: aplastic anemia) misclassified as trauma, and the two on the right (**B**) show traumatic cases (top right: abusive head trauma (AHT), bottom right: accidental trauma) wrongly predicted as medical.

**Figure 3 ijms-24-15105-f003:**
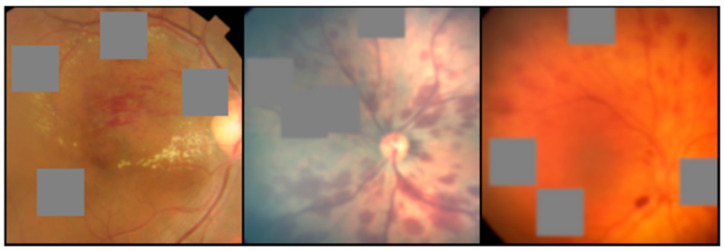
Depiction of the ‘cutout’ technique, which obscures random square regions to enhance the model’s focus diversity.

**Figure 4 ijms-24-15105-f004:**
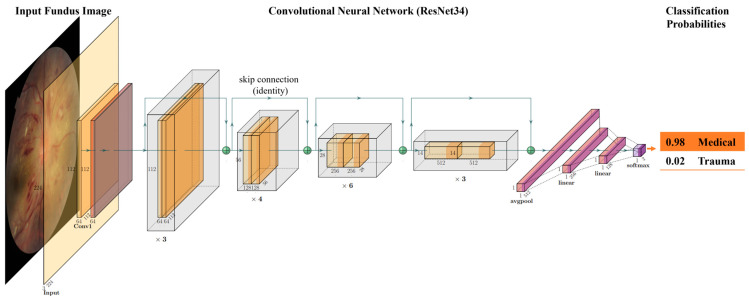
Visualization of the ResNet convolutional neural network (ResNet34) architecture and its application in our study. The flow diagram illustrates the journey from input fundus photos through the complex, multi-layered structure of the ResNet model, ending with the final task output.

**Table 1 ijms-24-15105-t001:** Dataset diagnosis characteristics.

Diagnosis	All (*n* = 599)	Training (*n* = 343)	Validation (*n* = 127)	Test (*n* = 127)
Medical	298	172	61	65
Retinal Vascular Disease	102	54	20	27
Leukemia	89	46	16	17
Papilledema	19	12	4	3
Coagulopathy	88	60	22	18
Trauma	299	171	66	62
Accidental Trauma	18	11	3	4
Birth Trauma	118	67	25	26
Abusive Head Trauma	163	93	38	32

**Table 2 ijms-24-15105-t002:** Performance metrics of the models.

Model	Accuracy	AUC	Specificity	Sensitivity	PPV	NPV
ResNet18	88.98%	0.9506	83.08%	**96.77%**	84.29%	**94.74%**
ResNet34	86.61%	0.9437	87.69%	87.10%	86.89%	86.36%
ResNet50	87.40%	0.9467	84.62%	91.94%	84.85%	90.16%
ResNet101	89.76%	0.9449	90.77%	90.32%	90.16%	89.39%
ResNet152	88.19%	0.9365	84.62%	93.55%	85.07%	91.67%
ResAttNet56	87.40%	0.9400	83.08%	93.55%	83.82%	91.53%
ViT-Small	79.53%	0.8945	78.46%	82.26%	78.12%	80.95%
FastViT-SA12	**90.55%**	**0.9628**	**96.92%**	85.48%	**96.30%**	86.30%
FastViT-SA24	88.19%	0.9462	87.69%	90.32%	87.30%	89.06%

AUC: Area under the receiver operating characteristic curve, PPV: Positive predictive value, NPV: Negative predictive value. Specificity, Sensitivity, PPV, and NPV were calculated at the Youden Index. Bolded values indicate the highest performance among the models.

**Table 3 ijms-24-15105-t003:** Model characteristics of the models.

Model	Number of Layers	Trainable Parameters (in Millions)
ResNet18	18	11.4
ResNet34	32	21.5
ResNet50	50	24.0
ResNet101	101	43.1
ResNet152	152	58.7
ResAttNet56	56	29.8
ViT-Small	12	22.5
FastVit-SA12	12	10.5
FastVit-SA24	24	20.5

## Data Availability

The data presented in this study are available upon request from the corresponding author. The data are not publicly available due to the sensitive nature of the images, which include fundus photos with retinal hemorrhages from Abusive Head Trauma patients.

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
