# Peer review of "Deep Learning Approach for Differentiating Etiologies of Pediatric Retinal Hemorrhages: A Multicenter Study"

_ijms, 2023, doi:10.3390/ijms242015105_

Round 1
Reviewer 1 Report
Thanks to the authors for sharing this study. here are my comments below.
1. The performance of experienced ophthalmologist should be revealed in the introduction. There is no way to compare the study result and the ophthalmologist. The discussion is unable to provide evidence of AI and ML performance comparable with human experts.
2. Specific techniques and performance of AI and ML from reviewed studies included in the introduction should be revealed so that the current status of ML in this area will be easier to understand.
3. Why does the study use small models of ResNet only? It is neither state-of-art nor has more layers. Updated models should be involved in this study
4. I presume that Table 2 is the result binary classification of the model to classify the classes of medical and trauma etiology. AOU graph and Confusion matrix should be provided.
5. The subclasses classification, related results and analysis are missing in this study.
6. The model fine-tuning strategy for hyperparameters is unclear. The optimizer is also missing.
Author Response
We thank the reviewers for their valuable input and thoughtful questions regarding our study. We appreciate their engagement with our research. We are resubmitting our revised manuscript. The manuscript has been carefully revised according to the reviewers’ comments as below.
Reviewer 1:
Thanks to the authors for sharing this study. here are my comments below.
Thank you for your thoughtful comments and suggestions. Your feedback has been instrumental in enhancing the comprehensiveness of our research.
- The performance of experienced ophthalmologist should be revealed in the introduction. There is no way to compare the study result and the ophthalmologist. The discussion is unable to provide evidence of AI and ML performance comparable with human experts.
Response: Thank you for pointing out this important aspect. It is true that comparing the performance of AI and ML algorithms with experienced ophthalmologists is challenging. While our study aims to explore the potential of computational tools in differentiating the etiologies of retinal hemorrhages, it is important to acknowledge the expertise of ophthalmologists in this field. We revised the introduction and discussion to highlight the role of experienced ophthalmologists as a benchmark and clarify that the goal is to complement their expertise with AI and ML technologies rather than replace it.
Introduction (Page 2, Paragraph 5):
“Unlike previous studies that primarily aimed to describe abnormal exams, our study seeks to develop and validate CNN models to differentiate between patients with known retinal hemorrhage. This is an especially challenging task as retinal hemorrhages can be difficult to interpret even for trained experts. One ROP study has shown that even the same expert can report different opinions on a retina image at different time points [23]. To date, despite the progress made in applying AI in ophthalmology, there is still a relative lack of research exploring the potential of computational tools in differentiating the etiologies of retinal hemorrhages.”
Discussion (Page 7, Paragraph 5):
"While our study aims to explore the potential of computational tools in differentiating the etiologies of retinal hemorrhages, it is important to acknowledge the expertise of experienced ophthalmologists as a benchmark in this field. The goal is to complement their expertise with AI and ML technologies, rather than replace it.”
- Specific techniques and performance of AI and ML from reviewed studies included in the introduction should be revealed so that the current status of ML in this area will be easier to understand.
Response: Thank you for your suggestion. Including specific techniques and performance metrics from reviewed studies will provide a clearer understanding of the current status of ML in this area. We revised the introduction to include relevant details about the techniques used and the performance achieved by AI and ML algorithms in differentiating retinal lesions and predicting disease progression.
Introduction (Page 2, Paragraph 4):
Studies have demonstrated that AI and ML algorithms, including deep learning models such as convolutional neural networks (CNNs), recurrent neural networks (RNNs), and transformers, can accurately identify and classify different types of retinal lesions, such as hemorrhages, exudates, and micro-aneurysms [25,26]. These algorithms can analyze large datasets of fundus photographs and provide automated assessments of disease severity and progression. Furthermore, AI and ML have been utilized to predict the risk of developing certain retinal diseases, such as geographic atrophy, based on fundus image analysis [27,29].
- Why does the study use small models of ResNet only? It is neither state-of-art nor has more layers. Updated models should be involved in this study
Thank you for suggesting to explore larger ResNet models, such as ResNet101 and ResNet154, as these models have demonstrated improved performance in various computer vision tasks. We performed additional experiments with these larger models to assess their efficacy for our specific task.
The results from the ResNet101 and ResNet154 models were added to our analysis. However, we found that, despite their increased complexity and depth, these larger models did not yield significantly better performance in terms of AUC when compared to the ResNet18 model. While the ResNet101 model showed slightly better accuracy by missing one less case, it did not exhibit a notable improvement in overall diagnostic performance.
To provide transparency and completeness, we have incorporated these results into Table 2 of our study. Additionally, the results and discussion sections have been updated to reflect these findings.
We believe that the inclusion of these results, which demonstrated the comparative performance of larger ResNet models, enriches the discussion and provides valuable insights into the model selection process for this specific medical diagnostic task.
Abstract (Page 1):
Deep learning models, specifically those based on ResNet architectures, were applied; ResNet101 achieved the highest accuracy, with a rate of 89.76%, while ResNet18 secured the highest area under the receiver operating characteristic curve (AUC) value, registering at 0.9506 on an independent test dataset.
Results (Page 6, section 2. Results, Paragraph 3):
Among the ResNet models, the ResNet18 model exhibited the most remarkable results, with an AUC of 0.9506 and an accuracy of 88.98%. It also demonstrated the highest sensitivity of 96.77% and a PPV of 84.29%. The ResNet101 model achieved an AUC of 0.9449, while attaining the highest accuracy, specificity, and PPV of 89.76%, 90.77%, and 90.16%, respectively out of the 5 models. Comprehensive performance metrics are presented in Table 2.
- I presume that Table 2 is the result binary classification of the model to classify the classes of medical and trauma etiology. AOU graph and Confusion matrix should be provided.
AUC graphs (Figure S1) for all models and Confusion matrices (Figure S2) for ResNet18 (highest AUC) and ResNet101 (highest accuracy) have been added to the supplementary.
- The subclasses classification, related results and analysis are missing in this study.
More details have been added to the results section. Number of misclassifications by diagnosis have been added to the Table S1 in the supplementary.
Results (Page 6, section 2. Results, Paragraph 3)
The ResNet101 model with the highest accuracy misclassified 7 medical and 6 trauma cases, while the ResNet18 model with the highest AUC misclassified 8 medical and 6 trauma cases. The misclassifications by diagnosis are presented in Table S1. It should be noted that the errors made by the models exhibited a discernible pattern …. Conversely, two instances of accidental and abusive head trauma, characterized by minimal retinal hemorrhage, were incorrectly identified as medical in origin. On the other hand, cases of papilledema and birth trauma were all classified correctly. Examples of these misclassifications are provided in Figure 2.
- The model fine-tuning strategy for hyperparameters is unclear. The optimizer is also missing.
As suggested, the manuscript is updated to include the optimizer and the hyperparameter tuning strategy for all ResNet models.
Methods (Page 8, Section 4.4 Algorithm Development, Paragraph 3):
Each model is trained for a total of 50 epochs. The model was optimized using Adam optimizer [45] with a standard binary cross-entropy loss function. We employed Hyperopt [46] for hyperparameter tuning and determined that a learning rate of 0.001 and a batch size of 32 consistently yielded the best performance across all ResNet models used in our study. Optimal model selection was determined based on validation data performance.
Reviewer 2 Report
I have a few comments/concerns regarding this section - 4.3. Data Preprocessing:
Authors stated this: "This involved resizing all images to a consistent shape of 256 × 256 pixels using bilinear interpolation, with zero-padding as necessary."
a) What was the original image size (fixed or non-fixed size)? Why was bilinear chosen over other interpolation algorithms? Why did the authors think that the dimension 256 x 256 is the most productive? While I agree with this image dimension size, I suggest authors refer to this work (see: https://www.mdpi.com/2079-9292/12/4/985) to better understand and get more ideas on why that was the best choice of image size dimension.
b) Regarding augmentation steps: Why "resizing the images to 224 × 224"? And why apply cropping at 250 with a scale range of 0.08 to 1? In addition, why was the maximum scale range set to 1?
c) Again, how did the authors handle the masks during data augmentation operation? For example, which interpolation method was used for mask processing?
Good
Author Response
I have a few comments/concerns regarding this section - 4.3. Data Preprocessing:
Thank you for your pertinent questions regarding the augmentation steps employed in our study. Your inquiry allows us to provide further clarity on our methodology.
Authors stated this: "This involved resizing all images to a consistent shape of 256 × 256 pixels using bilinear interpolation, with zero-padding as necessary."
- a) What was the original image size (fixed or non-fixed size)? Why was bilinear chosen over other interpolation algorithms? Why did the authors think that the dimension 256 x 256 is the most productive? While I agree with this image dimension size, I suggest authors refer to this work (see: https://www.mdpi.com/2079-9292/12/4/985) to better understand and get more ideas on why that was the best choice of image size dimension.
Thank you for including this insightful paper recommendation on image sizes. Unfortunately, we didn’t use segmentation masks in this task as we’re only doing a binary classification using only the fundus images as the input. We have updated the methods section to further explain our decision-making during training.
The original image sizes in our dataset varied due to the diverse sources and imaging equipment used in data collection. We chose to resize all images to a consistent shape of 256 × 256 pixels primarily for standardization, consistency, and compatibility of images from different centers with our deep learning models using pre-trained ResNet models for transfer learning.
Concerning the selection of bilinear interpolation over other interpolation algorithms, our decision was based on its suitability for our specific task. Bilinear interpolation strikes a balance between computational efficiency and preserving image information when resizing. While our dataset size may not be large, bilinear interpolation provides a reliable method for resizing images without introducing significant artifacts or distortions.
As for why we specifically chose the dimensions 256 x 256, this choice aligns with common practices in deep learning for image classification tasks. It strikes a balance between capturing essential features and keeping computational demands manageable. Additionally, it conforms to the requirements of many pre-trained models, allowing us to leverage transfer learning effectively.
Methods (Page 7, Section 4.3 Data Preprocessing, Paragraph 1):
This involved resizing all images to a consistent shape of 256 × 256 pixels using bilinear interpolation, with zero-padding as necessary. Bilinear interpolation provided a reliable method for resizing images without introducing significant artifacts or distortions.
- b) Regarding augmentation steps: Why "resizing the images to 224 × 224"? And why apply cropping at 250 with a scale range of 0.08 to 1? In addition, why was the maximum scale range set to 1?
The resizing of images to 224 × 224 was implemented as part of the data augmentation process for compatibility with the requirements of many pre-trained deep learning models, particularly those trained on ImageNet, which is a common practice in transfer learning. By using this standard size, we aimed to leverage pre-trained models effectively.
The choice of cropping with a scale range of 0.08 to 1 was made to introduce variability in the training data while ensuring that important information within the images is preserved. A minimum scale range of 0.08 ensures that even smaller details within the images are considered during training, potentially capturing subtle patterns that may be diagnostically relevant. The choice of a maximum scale range of 1 means that we allowed images to be cropped to their original size if needed. This ensures that the full image content is considered during training and helps maintain the integrity of the data.
Methods (Page 7, Section 4.3 Data Preprocessing, Paragraph 1):
… applying cropping with a scale range of 0.08 to 1. A minimum scale range of 0.08 ensures that even smaller details within the images are considered during training, potentially capturing subtle patterns that may be diagnostically relevant.
- c) Again, how did the authors handle the masks during data augmentation operation? For example, which interpolation method was used for mask processing?
There were no masks involved in the data augmentation process. We focused solely on augmenting the retinal fundus images themselves, without the use of masks or additional annotations.
Round 2
Reviewer 1 Report
Thanks for the effort in improvement. Here is my comments below.
1. State-of-art techniques and their performance in AI and ML from reviewed studies must be included, particularly attention-related.
2. The total number of augmented data should be revealed with respect to each augmented technique.
3. Attention mechanism is a common technique today. Suggestion to use attention resnet such as Residual Attention Network (attention-92) for studying and comparing with resnet. You may refer to the following paper.
Wang, F., Jiang, M., Qian, C., Yang, S., Li, C., Zhang, H., Wang, X., & Tang, X. (2017). Residual Attention Network for Image Classification. ArXiv. /abs/1704.06904
Author Response
We thank the reviewers for their valuable input on our revision. Please see that attached.

Round 3
Reviewer 1 Report
Thanks for the effort. The issues have been addressed.